# A Fair Comparison of Graph Neural Networks for Graph Classification

**Federico Errica**[*]
Department of Computer Science
University of Pisa
`federico.errica@phd.unipi.it`

**Marco Podda**[*]
Department of Computer Science
University of Pisa
`marco.podda@di.unipi.it`

**Davide Bacciu**[*]
Department of Computer Science
University of Pisa
`bacciu@di.unipi.it`

**Alessio Micheli**[*]
Department of Computer Science
University of Pisa
`micheli@di.unipi.it`

## Abstract

Experimental reproducibility and replicability are critical topics in machine learning. Authors have often raised concerns about their lack in scientific publications to improve the quality of the field. Recently, the graph representation learning field has attracted the attention of a wide research community, which resulted in a large stream of works. As such, several Graph Neural Network models have been developed to effectively tackle graph classification. However, experimental procedures often lack rigorousness and are hardly reproducible. Motivated by this, we provide an overview of common practices that should be avoided to fairly compare with the state of the art. To counter this troubling trend, we ran more than 47000 experiments in a controlled and uniform framework to re-evaluate five popular models across nine common benchmarks. Moreover, by comparing GNNs with structure-agnostic baselines we provide convincing evidence that, on some datasets, structural information has not been exploited yet. We believe that this work can contribute to the development of the graph learning field, by providing a much needed grounding for rigorous evaluations of graph classification models.

## 1 Introduction

Over the years, researchers have raised concerns about several flaws in scholarship, such as experimental reproducibility and replicability in machine learning (McDermott, 1976; Lipton & Steinhardt, 2018) and science in general (National Academies of Sciences & Medicine, 2019). These issues are not easy to address, as a collective effort is required to avoid bad practices. Examples include the ambiguity of experimental procedures, the impossibility of reproducing results and the improper comparison of machine learning models. As a result, it can be difficult to uniformly assess the effectiveness of one method against another. This work investigates these issues for the graph representation learning field, by providing a uniform and rigorous benchmarking of state-of-the-art models.

Graph Neural Networks (GNNs) (Micheli, 2009; Scarselli et al., 2008) have recently become the standard tool for machine learning on graphs. These architectures effectively combine node features and graph topology to build distributed node representations. GNNs can be used to solve node classification (Kipf & Welling, 2017) and link prediction (Zhang & Chen, 2018) tasks, or they can be applied to downstream graph classification (Bacciu et al., 2018). In literature, such models are usually evaluated on chemical and social domains (Xu et al., 2019).
Given their appeal, an ever increasing number of GNNs is being developed (Gilmer et al., 2017). However, despite the theoretical advancements reached by the latest contributions in the field, we find that the experimental settings are in many cases ambiguous or not reproducible.

---

[*]Equal contribution.

Some of the most common reproducibility problems we encounter in this field concern hyper-parameters selection and the correct usage of data splits for model selection versus model assessment. Moreover, the evaluation code is sometimes missing or incomplete, and experiments are not standardized across different works in terms of node and edge features.

These issues easily generate doubts and confusion among practitioners that need a fully transparent and reproducible experimental setting. As a matter of fact, the evaluation of a model goes through two different phases, namely *model selection* on the validation set and *model assessment* on the test set. Clearly, to fail in keeping these phases well separated could lead to over-optimistic and biased estimates of the true performance of a model, making it hard for other researchers to present competitive results without following the same ambiguous evaluation procedures.

With this premise, our primary contribution is to provide the graph learning community with a fair performance comparison among GNN architectures, using a standardized and reproducible experimental environment. More in detail, we performed a large number of experiments within a rigorous model selection and assessment framework, in which all models were compared using the same features and the same data splits.

Secondly, we investigate if and to what extent current GNN models can effectively exploit graph structure. To this end, we add two domain-specific and structure-agnostic baselines, whose purpose is to disentangle the contribution of structural information from node features. Much to our surprise, we found out that these baselines can even perform better than GNNs on some datasets; this calls for moderation when reporting improvements that do not clearly outperform structure-agnostic competitors.

Our last contribution is a study on the effect of node degrees as features in social datasets. Indeed, we show that providing the degree can be beneficial in terms of performances, and it has also implications in the number of GNN layers needed to reach good results.

We publicly release code and dataset splits to reproduce our results, in order to allow other researchers to carry out rigorous evaluations with minimum additional effort[1].

**Disclaimer**    Before delving into the work, we would like to clarify that this work does *not* aim at pinpointing the best (or worst) performing GNN, nor it disavows the effort researchers have put in the development of these models. Rather, it is intended to be an attempt to set up a standardized and uniform evaluation framework for GNNs, such that future contributions can be compared fairly and objectively with existing architectures.

## 2    RELATED WORK

**Graph Neural Networks**    At the core of GNNs is the idea to compute a state for each node in a graph, which is iteratively updated according to the state of neighboring nodes. Thanks to layering (Micheli, 2009) or recursive (Scarselli et al., 2008) schemes, these models propagate information and construct node representations that can be "aware" of the broader graph structure. GNNs have recently gained popularity because they can efficiently and automatically extract relevant features from a graph; in the past, the most popular way to deal with complex structures was to use kernel functions (Shervashidze et al., 2011) to compute task-agnostic features. However, such kernels are non-adaptive and typically computationally expensive, which makes GNNs even more appealing.

Even though in this work we specifically focus on architectures designed for graph classification, all GNNs share the notion of "convolution" over node neighborhoods, as a generalization of convolution on grids. For example, GraphSAGE (Hamilton et al., 2017) first performs sum, mean or max-pooling neighborhood aggregation, and then it updates the node representation applying a linear projection on top of the convolution. It also relies on a neighborhood *sampling* scheme to keep computational complexity constant. Instead, Graph Isomorphism Network (GIN) (Xu et al., 2019) builds upon the limitations of GraphSAGE, extending it with arbitrary aggregation functions on multi-sets. The model is proven to be as theoretically powerful as the Weisfeiler-Lehman test of graph isomorphism. Very recently, Wagstaff et al. (2019) gave an upper bound to the number of hidden units needed to learn permutation-invariant functions over sets and multi-sets. Differently from the above methods, Edge-Conditioned Convolution (ECC) (Simonovsky & Komodakis, 2017)

---

[1]Code available at: `https://github.com/diningphil/gnn-comparison`

learns a different parameter for each edge label. Therefore, neighbor aggregation is weighted according to specific edge parameters. Finally, Deep Graph Convolutional Neural Network (DGCNN) (Zhang et al., 2018) proposes a convolutional layer similar to the formulation of Kipf & Welling (2017).

Some models also exploit a pooling scheme, which is applied after convolutional layers in order to reduce the size of a graph. For example, the pooling scheme of ECC coarsens graphs through a differentiable pooling map that can be pre-computed. Similarly, DiffPool (Ying et al., 2018) proposes an adaptive pooling mechanism that collapses nodes on the basis of a supervised criterion. In practice, DiffPool combines a differentiable graph encoder with its pooling strategy, so that the architecture is end-to-end trainable. Lastly, DGCNN differs from other works in that nodes are sorted and aligned by a specific algorithm called SortPool (Zhang et al., 2018).

**Model evaluation** The work of Shchur et al. (2018) shares a similar purpose with our contribution. In particular, the authors compare different GNNs on node classification tasks, showing that results are highly dependent on the particular train/validation/test split of choice, up to the point where changing splits leads to dramatically different performance rankings. Thus, they recommend to evaluate GNNs on multiple test splits to achieve a fair comparison. Even though we operate in a different setting (graph instead of node classification), we follow the authors' suggestions by evaluating models under a controlled and rigorous assessment framework. Finally, the work of Dacrema et al. (2019) criticizes a large number of neural recommender systems, most of which are not reproducible, showing that only one of them truly improves against a simple baseline.

## 3 RISK ASSESSMENT AND MODEL SELECTION

Here, we recap the risk assessment (also called model evaluation or model assessment) and model selection procedures, to clearly layout the experimental procedure followed in this paper. For space reasons, the overall procedure is visually summarized in Appendix A.1.

### 3.1 RISK ASSESSMENT

The goal of risk assessment is to provide an estimate of the performance of a class of models. When a test set is not explicitly given, a common way to proceed is to use *k-fold Cross Validation* (CV) (Stone, 1974; Varma & Simon, 2006; Cawley & Talbot, 2010). $k$-fold CV uses $k$ different training/test splits to estimate the generalization performance of a model; for each partition, an internal model selection procedure selects the hyper-parameters using the training data only. This way, test data is **never** used for model selection. As model selection is performed independently for each training/test split, we obtain different "best" hyper-parameter configurations; this is why we refer to the performance of a class of models.

### 3.2 MODEL SELECTION

The goal of model selection, or hyper-parameter tuning, is to choose among a set of candidate hyper-parameter configurations the one that works best on a specific *validation* set. If a validation set is not given, one can rely on a holdout training/validation split or an inner $k$-fold. Nevertheless, the key point to remember is that validation performances are *biased* estimates of the true generalization capabilities. Consequently, model selection results are generally over-optimistic; this issue is thoroughly documented in Cawley & Talbot (2010). This is why the main contribution of this work is to clearly separate model selection and model assessment estimates, something that is lacking or ambiguous in the literature under consideration.

## 4 OVERVIEW OF REPRODUCIBILITY ISSUES

To motivate our contribution, we follow the approach of Dacrema et al. (2019) and briefly review recent papers describing five different GNN models, highlighting problems in the experimental setups as well as reproducibility of results. We emphasize that our observations are based solely on the contents of their paper and the available code[2]. Suitable GNN works were selected according to

---

[2]As of the date of this submission.

the following criteria: i) performances obtained with 10-fold CV; ii) peer reviewed; iii) strong architectural differences; iv) popularity. In particular, we selected DGCNN (Zhang et al., 2018), DiffPool (Ying et al., 2018), ECC (Simonovsky & Komodakis, 2017), GIN (Xu et al., 2019) and GraphSAGE (Hamilton et al., 2017). For a detailed description of each model we refer to their respective papers. Our criteria to assess quality of evaluation and reproducibility are: *i*) code for data preprocessing, model selection and assessment is provided; *ii*) data splits are provided; *iii*) data is split by means of a stratification technique, to preserve class proportions across all partitions; *iv*) results of the 10-fold CV are reported correctly using standard deviations, and they refer to model evaluation (test sets) rather than model selection (validation sets). Table 1 summarizes our findings.

Table 1: Criteria for reproducibility considered in this work and their compliance among considered models. (Y) indicates that the criterion is met, (N) indicates that the criterion is not satisfied, (A) indicates ambiguity (i.e. it is unclear whether the criteria is met or not), (-) indicates lack of information (i.e. no details are provided about the criteria). Note that GraphSAGE is excluded from this comparison, as it was not directly applied by authors to graph classification tasks.

|  | DGCNN | DiffPool | ECC | GIN |
|---|---|---|---|---|
| Data preprocessing code | Y | Y | - | Y |
| Model selection code | N | N | - | N |
| Model evaluation code | Y | Y | - | Y |
| Data splits provided | Y | N | N | Y |
| Label Stratification | Y | N | - | Y |
| Report accuracy on test | Y | A | A | N |
| Report standard deviations | Y | N | N | Y |

**DGCNN** The authors evaluate the model on 10-fold CV. While the architecture is fixed for all dataset, learning rate and epochs are tuned using only one random CV fold, and then reused on all the other folds. While this practice is still acceptable, it may lead to sub-optimal performances. Nonetheless, the code to reproduce model selection is not available. Moreover, the authors run CV 10 times, and they report the average of the 10 final scores. As a result, the variance of the provided estimates is reduced. However, the same procedure was not applied to the other competitors as well. Finally, CV data splits are correctly stratified and publicly available, making it possible to reproduce at least the evaluation experiments.

**DiffPool** From both the paper and the provided code, it is unclear if reported results are obtained on a test set rather than a validation set. Although the authors state that 10-fold CV is used, standard deviations of DiffPool and its competitors are not reported. Moreover, the authors affirm to have applied early stopping on the validation set to prevent overfitting; unfortunately, neither model selection code nor validation splits are available. Furthermore, according to the code, data is randomly split (without stratification) and no random seed is set, hence splits are different each time the code is executed.

**ECC** The paper reports that ECC is evaluated on 10-fold CV, but results do not include standard deviations. Similarly to DGCNN, hyper-parameters are fixed in advance, hence it is not clear if and how model selection has been performed. Importantly, there are no references in the code repository to data pre-processing, data stratification, data splitting, and model selection.

**GIN** The authors correctly list all the hyper-parameters tuned. However, as stated explicitly in the paper and in the public review discussion, they report the *validation* accuracy of 10-fold CV. In other words, reported results refer to model selection and not to model evaluation. The code for model selection is not provided.

**GraphSAGE** The original paper does not test this model on graph classification datasets, but GraphSAGE is often used in other papers as a strong baseline. It follows that GraphSAGE results on graph classification should be accompanied by the code to reproduce the experiments. Despite that, the two works which report results of GraphSAGE (DiffPool and GIN) fail to do so.

**Summary**   Our analysis reveals that GNN works rarely comply with good machine learning practices as regards the quality of evaluation and reproducibility of results. This motivates the need to re-evaluate all models within a rigorous, reproducible and fair environment.

## 5   EXPERIMENTS

In this section we detail our main experiment, in which we re-evaluate the above-mentioned models on 9 datasets (4 chemical, 5 social), using a model selection and assessment framework that closely follows the rigorous practices described in Section 3. In addition, we implement two baselines whose purpose is to understand the extent to which GNNs are able to exploit structural information. All models have been implemented by means of the Pytorch Geometrics library (Fey & Lenssen, 2019), which provides graph pre-processing routines and makes the definition of graph convolution easier to implement. We sometimes found discrepancies between papers and related code; in such cases, we complied with the specifications in the paper. Because GraphSAGE was not applied to graph classification in the original work, we opted for a max-pooling global aggregation function to classify graph instances; further, we do not use the sampled neighborhood aggregation scheme defined in Hamilton et al. (2017), in order to allow nodes to have access to their whole neighborhood.

**Datasets**   All graph datasets are publicly available (Kersting et al., 2016) and represent a relevant subset of those most frequently used in literature to compare GNNs. Some collect molecular graphs, while others contain social graphs. In particular, we used D&D (Dobson & Doig, 2003), PROTEINS (Borgwardt et al., 2005), NCI1 (Wale et al., 2008) and ENZYMES (Schomburg et al., 2004) for binary and multi-class classification of chemical compounds, whereas IMDB-BINARY, IMDB-MULTI, REDDIT-BINARY, REDDIT-5K and COLLAB (Yanardag & Vishwanathan, 2015) are social datasets. Dataset statistics are reported in Table A.2.

**Features**   In GNN literature, it is common practice to augment node descriptors with structural features. For example, DiffPool adds the degree and clustering coefficient to each node feature vector, whereas GIN adds a one-hot representation of node degrees. The latter choice trades off an improvement in performances (due to injectivity of the first sum) with the inability to generalize to graphs with arbitrary node degree.
In general, good experimental practices suggest that all models should be consistently compared to the same input representations. This is why we re-evaluate *all* models using the same node features. In particular, we use one common setting for the chemical domain and two alternative settings as regards the social domain. As regards the chemical domain, nodes are labeled with a one-hot encoding of their atom type, though on ENZYMES we follow the literature and use 18 additional features available. As regards social graphs, whose nodes do not have features, we use either an uninformative feature for all nodes or the node degree. As such, we are able to reason about the effectiveness of the structural inductive bias imposed by the model; that is if the model is able to implicitly learn structural features or not. The effect of adding structural features to general machine learning models for graphs has been investigated in Gallagher & Eliassi-Rad (2008); here, we focus on the impact of node degrees on performances for social datasets.

**Baselines**   We adopt two distinct baselines, one for chemical and one for social datasets. On all chemical datasets but for ENZYMES, we follow Ralaivola et al. (2005); Luzhnica et al. (2019) and implement the Molecular Fingerprint technique, which first applies global sum pooling (i.e., counts the occurrences of atom types in the graph by summing the features of *all nodes* in the graph together) and then applies a single-layer MLP with ReLU activations. On social domains and ENZYMES (due to the presence of additional features), we take inspiration from the work of Zaheer et al. (2017) to learn permutation-invariant functions over sets of nodes: first, we apply a single-layer MLP on top of node features, followed by global sum pooling and another single-layer MLP for classification. Note that both baselines do not leverage graph topology. Using these baselines as a reference is of fundamental importance for future works, as they can provide feedback on the effectiveness of GNNs on a specific dataset. As a matter of fact, if GNN performances are close to the ones of a structure-agnostic baseline, one can draw two possible conclusions: the task does not need topological information to be effectively solved, or the GNN is not exploiting graph structure adequately. While the former can be verified through domain-specific human expertise, the second is more difficult to assess, as multiple factors come into play such as the amount of training

data, the structural inductive bias imposed by the architecture and the hyper-parameters used for model selection. Nevertheless, *significant* improvements with respect to these baselines are a strong indicator that graph topology has been exploited. Therefore, structure-agnostic baselines become vital to understand if and how a model can be improved.

Table 2: Pseudo-code for model assessment (left) and model selection (right). In Algorithm 1, "Select" refers to Algorithm 2, whereas "Train" and "Eval" represent training and inference phases, respectively. After each model selection, the best configuration $best_k$ is used to evaluate the external test fold. Performances are averaged across $R$ training runs, where $R$ in our case is set to 3.

---

**Algorithm 1** Model Assessment ($k$-fold CV)

1: Input: Dataset $\mathcal{D}$, set of configurations $\Theta$
2: Split $\mathcal{D}$ into $k$ folds $F_1, \ldots, F_k$
3: **for** $i \leftarrow 1, \ldots, k$ **do**
4:     $train_k, test_k \leftarrow \left( \bigcup_{j \neq i} F_j \right), F_i$
5:     $best_k \leftarrow \text{Select}(train_k, \Theta)$
6:     **for** $r \leftarrow 1, \ldots, R$ **do**
7:         $model_r \leftarrow \text{Train}(train_k, best_k)$
8:         $p_r \leftarrow \text{Eval}(model_k, test_k)$
9:     **end for**
10:    $perf_k \leftarrow \sum_{r=1}^{R} p_r / R$
11: **end for**
12: **return** $\sum_{i=1}^{k} perf_i / k$

---

**Algorithm 2** Model Selection

1: Input: $train_k, \Theta$
2: Split $train_k$ into *train* and *valid*
3: $p_\theta = \emptyset$
4: **for each** $\theta \in \Theta$ **do**
5:     $model \leftarrow \text{Train}(train_k, \theta)$
6:     $p_\theta \leftarrow p_\theta \cup \text{Eval}(model, valid)$
7: **end for**
8: $best_\theta \leftarrow \text{argmax}_\theta \, p_\theta$
9: **return** $best_\theta$

---

**Experimental Setting**  Our experimental approach is to use a 10-fold CV for model assessment and an inner holdout technique with a 90%/10% training/validation split for model selection. After *each* model selection, we train three times on the whole training fold, holding out a random fraction (10%) of the data to perform early stopping. These three separate runs are needed to smooth the effect of unfavorable random weight initialization on test performances. The final test fold score is obtained as the mean of these three runs; Table 2 reports the pseudo-code of the entire evaluation process. To be consistent with literature, we implement early stopping with patience parameter $n$, where training stops if $n$ epochs have passed without improvement on the validation set. A high value of $n$ can favor model selection by making it less sensitive to fluctuations in the validation score at the cost of additional computation. Importantly, all data partitions have been pre-computed, so that models are selected and evaluated on the same data splits. Moreover, all data splits are stratified, i.e., class proportions are preserved inside each $k$-fold split as well as in the holdout splits used for model selection.

**Hyper-parameters**  Hyper-parameter tuning is performed via grid search. For the sake of conciseness, we list all hyper-parameters in Section A.4. Notice that we always include those used by other authors in their respective papers. We select the number of convolutional layers, the embedding space dimension, the learning rate, and the criterion for early stopping (either based on the validation accuracy or validation loss) for all models. Depending on the model, we also selected regularization terms, dropout, and other model-specific parameters.

**Computational considerations**  Our experiments involve a large number of training runs. For all models, grid sizes range from 32 to 72 possible configurations, depending on the number of hyper-parameters to choose from. However, we tried to keep the upper bound on the number of parameters as similar as possible across models. The total effort required, in terms of the number of single training runs, to complete model assessment procedures exceeded 47000. Such a large number required extensive use of parallelism, both in CPU and GPU, to conduct the experiments in a reasonable amount of time. We emphasize that in some cases (e.g. ECC in social datasets), training on a *single* hyper-parameter configuration required more than 72 hours, which would have made the sequential exploration of one single grid last months. Therefore, due to the large amount of experiments to conduct and to the computational resources available, we limited the time to complete a single training to 72 hours.

Table 3: Results on chemical datasets with mean accuracy and standard deviation are reported. Best performances are highlighted in bold.

|  | D&D | NCI1 | PROTEINS | ENZYMES |
|---|---|---|---|---|
| Baseline | **78.4** ± 4.5 | 69.8 ± 2.2 | **75.8** ± 3.7 | **65.2** ± 6.4 |
| DGCNN | 76.6 ± 4.3 | 76.4 ± 1.7 | 72.9 ± 3.5 | 38.9 ± 5.7 |
| DiffPool | 75.0 ± 3.5 | 76.9 ± 1.9 | 73.7 ± 3.5 | 59.5 ± 5.6 |
| ECC | 72.6 ± 4.1 | 76.2 ± 1.4 | 72.3 ± 3.4 | 29.5 ± 8.2 |
| GIN | 75.3 ± 2.9 | **80.0** ± 1.4 | 73.3 ± 4.0 | 59.6 ± 4.5 |
| GraphSAGE | 72.9 ± 2.0 | 76.0 ± 1.8 | 73.0 ± 4.5 | 58.2 ± 6.0 |

# 6 RESULTS AND DISCUSSION

Tables 3 and 4 show the results of our experiments. Overall, GIN seems to be effective on social datasets. Importantly, we discover that on D&D, PROTEINS and ENZYMES none of the GNNs are able to improve over the baseline. On the contrary, on NCI1 the baseline is clearly outperformed: this result suggests that the GNNs we analyzed can actually exploit the topological information of the graphs in this dataset. Moreover, we observe that an overly-parameterized baseline is not able to overfit the NCI1 training data completely. To see this, consider that a baseline with 10000 hidden units and no regularization reaches around 67% training accuracy, while GIN can easily overfit ($\approx 100\%$) the training data. This indicates that structural information hugely affects the ability to fit the training set. On social datasets, we observe that adding node degrees as features is beneficial, but such an effect is more noticeable for REDDIT-BINARY, REDDIT-5K and COLLAB.

## 6.1 THE IMPORTANCE OF BASELINES

Our results also show that structure-agnostic baselines are an essential tool to understand the effectiveness of GNNs and extract useful insights. As an example, since none of the GNNs surpasses the baseline on D&D, PROTEINS and ENZYMES, we argue that the state-of-the-art GNN models we analyzed are not able to fully exploit the structure on such datasets yet; indeed, in chemistry, structural features are known to correlate with molecular properties (van Rossum, 1963). For all these reasons, we suggest putting small performance gains on these datasets into the right perspective, at least until the baseline will clearly be outperformed. Currently, small average fluctuations on these datasets are likely to be caused by other factors, such as random initializations, rather than a successful exploitation of the structure. In conclusion, we warmly recommend GNN practitioners to include baseline comparisons in future works, in order to better characterize the extent of their contributions.

## 6.2 THE EFFECT OF NODE DEGREE

Based on our results, using node degrees as input features is almost always beneficial to increase performances on social datasets, sometimes by a large amount. As an example, degree information is sufficient for our baseline to improve performances of $\approx 15\%$, hence being competitive on many datasets; in particular, the baseline achieves the best performance on IMDB-BINARY. In contrast, adding node degrees is less relevant for most GNNs, since they can automatically infer such information from the structure. One notable exception is DGCNN, which explicitly needs node degrees to perform well on all datasets. Moreover, we observe that the ranking of all models, after the addition of the degrees, drastically changes; this raises the question about the impact of other structural features (such as clustering coefficient) on performances, which we leave to future works.

However, one may also wonder whether the addition of the degree has an influence on the number of layers that are necessary to solve the task or not. We therefore investigated the matter by computing the median number of layers across the 10 different folds. We observed a general trend across models, with GraphSAGE being the only exception, where the addition of the degree reduces the number of layers needed by $\approx 1$ as shown in Table A.3. This may be due to the fact that most architectures find useful to compute the degree at the very first layer, as such information seems useful to the overall performances.

Table 4: Results on social datasets with mean accuracy and standard deviation are reported. Best performances are highlighted in bold. OOR means Out of Resources, either time (> 72 hours for a single training) or GPU memory.

| | | IMDB-B | IMDB-M | REDDIT-B | REDDIT-5K | COLLAB |
|---|---|---|---|---|---|---|
| No Features | Baseline | $50.7 \pm 2.4$ | $36.1 \pm 3.0$ | $72.1 \pm 7.8$ | $35.1 \pm 1.4$ | $55.0 \pm 1.9$ |
| | DGCNN | $53.3 \pm 5.0$ | $38.6 \pm 2.2$ | $77.1 \pm 2.9$ | $35.7 \pm 1.8$ | $57.4 \pm 1.9$ |
| | DiffPool | $68.3 \pm 6.1$ | $45.1 \pm 3.2$ | $76.6 \pm 2.4$ | $34.6 \pm 2.0$ | $67.7 \pm 1.9$ |
| | ECC | $67.8 \pm 4.8$ | $44.8 \pm 3.1$ | OOR | OOR | OOR |
| | GIN | $66.8 \pm 3.9$ | $42.2 \pm 4.6$ | $\mathbf{87.0} \pm 4.4$ | $\mathbf{53.8} \pm 5.9$ | $\mathbf{75.9} \pm 1.9$ |
| | GraphSAGE | $\mathbf{69.9} \pm 4.6$ | $\mathbf{47.2} \pm 3.6$ | $86.1 \pm 2.0$ | $49.9 \pm 1.7$ | $71.6 \pm 1.5$ |
| With Degree | Baseline | $70.8 \pm 5.0$ | $\mathbf{49.1} \pm 3.5$ | $82.2 \pm 3.0$ | $52.2 \pm 1.5$ | $70.2 \pm 1.5$ |
| | DGCNN | $69.2 \pm 3.0$ | $45.6 \pm 3.4$ | $87.8 \pm 2.5$ | $49.2 \pm 1.2$ | $71.2 \pm 1.9$ |
| | DiffPool | $68.4 \pm 3.3$ | $45.6 \pm 3.4$ | $89.1 \pm 1.6$ | $53.8 \pm 1.4$ | $68.9 \pm 2.0$ |
| | ECC | $67.7 \pm 2.8$ | $43.5 \pm 3.1$ | OOR | OOR | OOR |
| | GIN | $\mathbf{71.2} \pm 3.9$ | $48.5 \pm 3.3$ | $\mathbf{89.9} \pm 1.9$ | $\mathbf{56.1} \pm 1.7$ | $\mathbf{75.6} \pm 2.3$ |
| | GraphSAGE | $68.8 \pm 4.5$ | $47.6 \pm 3.5$ | $84.3 \pm 1.9$ | $50.0 \pm 1.3$ | $73.9 \pm 1.7$ |

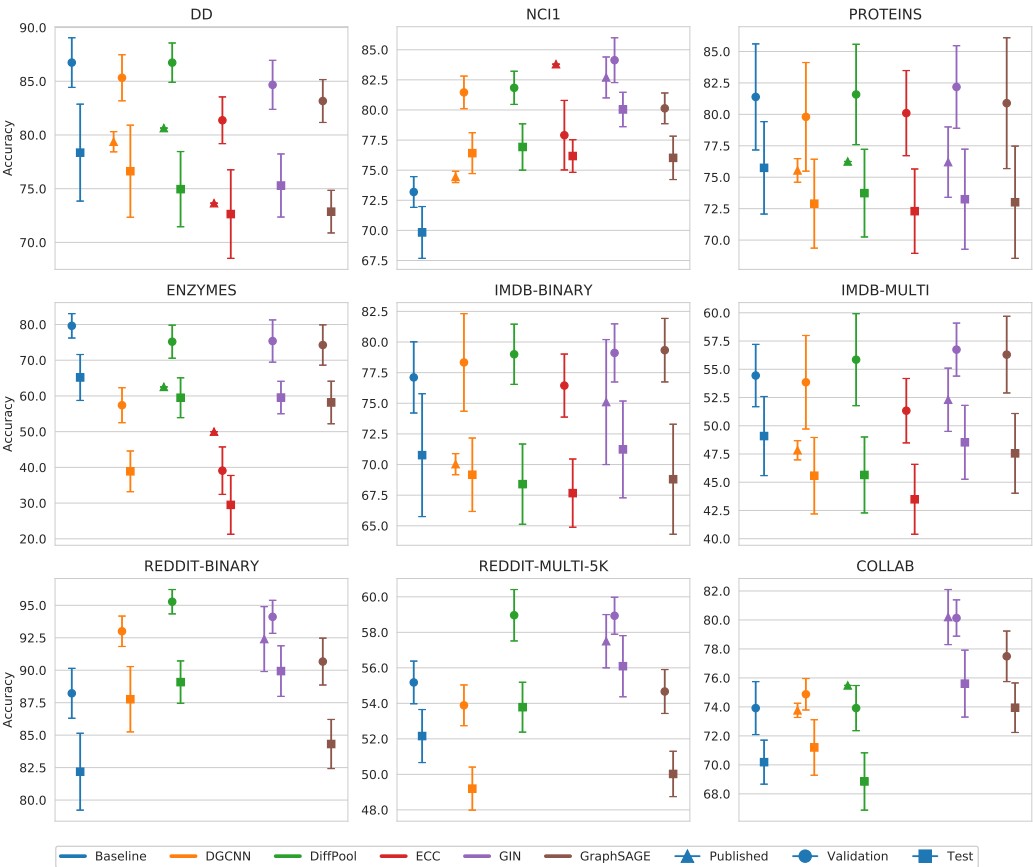

Figure 1: Chemical and social (with degree) benchmark results are shown together with published results (when available). For each of them, we report validation and test accuracies of the evaluated models, together with published results if available.

## 6.3 COMPARISON WITH PUBLISHED RESULTS

Figure 1 compares the average values of our test results with those reported in literature. In addition, we plot the average of our validation results across the 10 different model selections. The plots show how our test accuracies are in most cases different from what reported in the literature, and the gap

between the two estimates is usually consistent. In contrast, our average validation accuracies are always higher or equal to our test results; this is expected, as discussed in Section 3.2. Finally, we emphasize once again that our results are *i*) obtained within the framework of a rigorous model selection and assessment protocol; *ii*) fair with respect of data splits and input features assigned to all competitors; *iii*) reproducible. In contrast, we saw in Section 4 how published results rely on unclear or poorly documented experimental settings.

## 7 CONCLUSIONS

In this paper, we wanted to show how a rigorous empirical evaluation of GNNs can help design future experiments and better reason about the effectiveness of different architectural choices. To this aim, we highlighted ambiguities in the experimental settings of different papers, and we proposed a clear and reproducible procedure for future comparisons. We then provided a complete re-evaluation of five GNNs on nine datasets, which required a significant amount of time and computational resources. This uniform environment helped us reason about the role of structure, as we found that structure-agnostic baselines outperform GNNs on some chemical datasets, thus suggesting that structural properties have not been exploited yet. Moreover, we objectively analyzed the effect of the degree feature on performances and model selection in social datasets, unveiling an effect on the depth of GNNs. Finally, we provide the graph learning community with reliable and reproducible results to which GNN practitioners can compare their architectures. We hope that this work, along with the library we release, will prove useful to researchers and practitioners that want to compare GNNs in a more rigorous way.

ACKNOWLEDGMENTS

D. Bacciu would like to acknowledge support from the Italian Ministry of Education, University, and Research (MIUR) under project SIR 2014 LIST-IT (grant n. RBSI14STDE).

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

## A APPENDIX

### A.1 VISUALIZATION OF THE EVALUATION FRAMEWORK

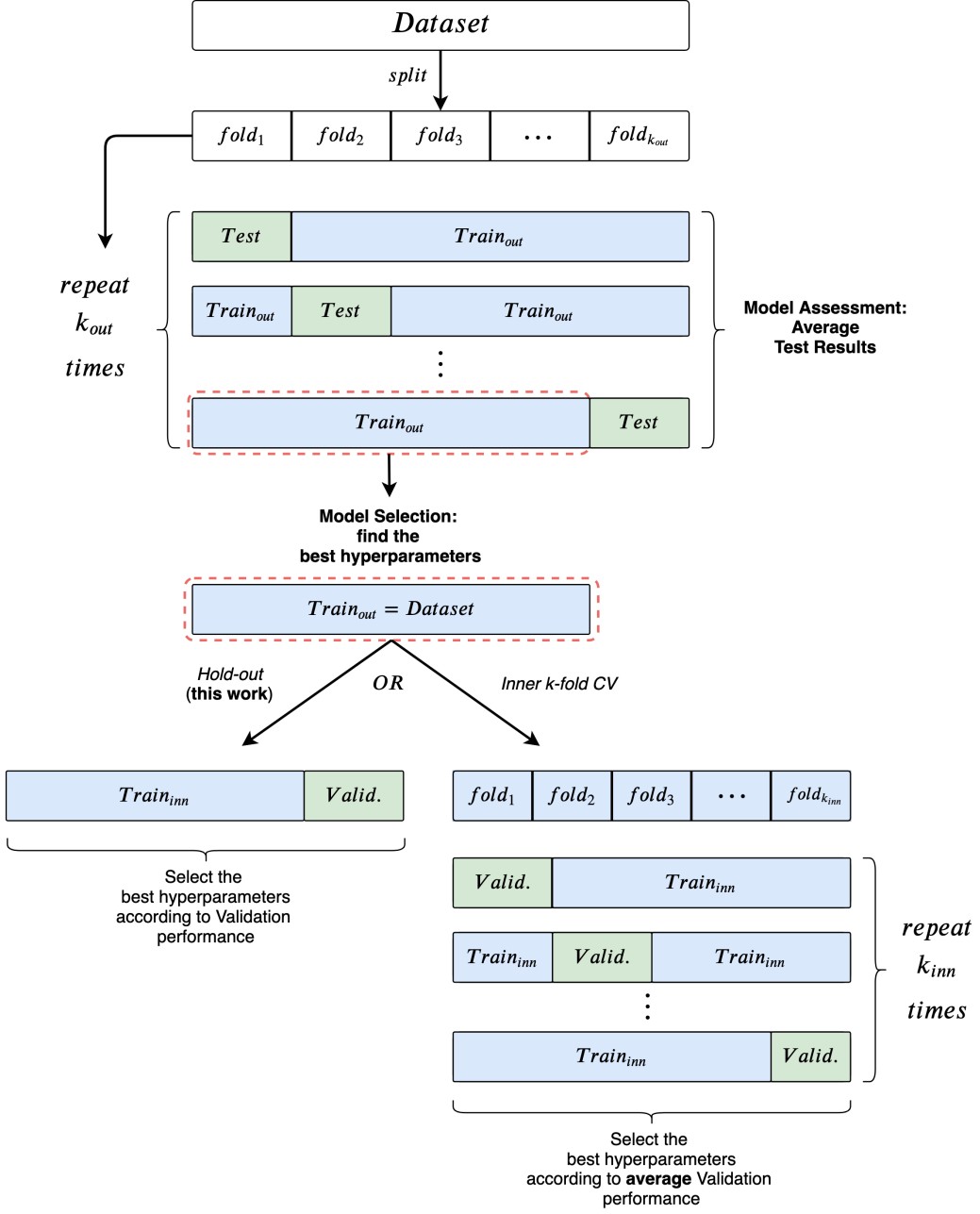

Figure 2: We give a visual representation of the evaluation framework. We apply an external $k_{out}$-fold CV to get an estimate of the generalization performance of a model, and we use an hold-out technique (bottom-left) to select the best hyper-parametres. For completeness, we show that it is also possible to apply an inner $k_{inn}$-fold CV (implementing a complete *Nested Cross Validation*), which obviously amounts to multiplying the computational costs of model selection by a factor $k_{inn}$.

## A.2 DATASET STATISTICS

Table 5: Dataset Statistics. Note that, when node labels are not present, we either assigned the same feature of 1 or the degree to all nodes in the dataset. Moreover, following the literature, we use the 18 additional node attributes for ENZYMES.

|  |  | # Graphs | # Classes | # Nodes | # Edges | # Node labels |
|---|---|---|---|---|---|---|
| CHEM. | DD | 1178 | 2 | 284.32 | 715.66 | 89 |
|  | ENZYMES | 600 | 6 | 32.63 | 64.14 | 3 |
|  | NCI1 | 4110 | 2 | 29.87 | 32.30 | 37 |
|  | PROTEINS | 1113 | 2 | 39.06 | 72.82 | 3 |
| SOCIAL | COLLAB | 5000 | 3 | 74.49 | 2457.78 | - |
|  | IMDB-BINARY | 1000 | 2 | 19.77 | 96.53 | - |
|  | IMDB-MULTI | 1500 | 3 | 13.00 | 65.94 | - |
|  | REDDIT-BINARY | 2000 | 2 | 429.63 | 497.75 | - |
|  | REDDIT-5K | 4999 | 5 | 508.82 | 594.87 | - |

## A.3 EFFECT OF NODE DEGREE ON LAYERING

Table 6: The table displays the median number of selcted layers in relation to the addition of node degrees as input features on all social datasets. 1 indicates that an uninformative feature is used as node label.

|  | IMDB-B | | IMDB-M | | REDDIT-B | | REDDIT-M | | COLLAB | |
|---|---|---|---|---|---|---|---|---|---|---|
|  | 1 | DEG | 1 | DEG | 1 | DEG | 1 | DEG | 1 | DEG |
| **DGCNN** | 3 | 3 | 3.5 | 3 | 4 | 3 | 3 | 2 | 4 | 2 |
| **DiffPool** | 1 | 2 | 2 | 1 | 2 | 2 | 2 | 1 | 2 | 1.5 |
| **ECC** | 1 | 2 | 1 | 1 | - | - | - | - | - | - |
| **GIN** | 3 | 2 | 4 | 2 | 4 | 4 | 4 | 3 | 4 | 4 |
| **GraphSAGE** | 4 | 3 | 5 | 4 | 3 | 4 | 3 | 5 | 3 | 5 |

## A.4 HYPER-PARAMETERS TABLE

Table 7: Hyper-parameters used for model selection.

| | Layers | Convs per layer | Batch size | Learning rate | Hidden units | Epochs | L2 | Dropout | Patience | Optimizer | Scheduler | Dense dim | Embed. dim | Neighbors Aggregation |
|---|---|---|---|---|---|---|---|---|---|---|---|---|---|---|
| Baseline chemical | - | - | 32
128 | 1e-1
1e-3
1e-6 | 32
128
256 | 5000 | 1e-2
1e-3
1e-4 | - | 500, loss
500, acc | Adam | - | - | - | sum |
| Baseline IMDB | - | - | 32
128 | 1e-1
1e-3
1e-6 | 32
128
256 | 3000 | 1e-2
1e-3
1e-4 | - | 500, loss
500, acc | Adam | - | - | - | sum |
| Base. COLLAB and REDDIT | - | - | 32
128 | 1e-1
1e-3 | 32
128 | 3000 | 1e-2
1e-3
1e-4 | - | 500, loss
500, acc | Adam | - | - | - | sum |
| Baseline ENZYMES | - | - | 32 | 1e-1
1e-3
1e-6 | 32
64
128
256 | 5000 | 1e-2
1e-3
1e-4 | - | 1000, loss
1000, acc | Adam | - | - | - | sum |
| DGCNN | 2
3
4 | 1 | 50 (cpu)
16 (gpu) | 1e-4
1e-5 | 32
64 | 1000 | - | 0.5 | 500, loss
500, acc | Adam | - | 128 | - | mean |
| DiffPool | 1
2 | 3 | 20 (cpu)
8 (gpu) | 1e-3
1e-4
1e-5 | 32
64 | 3000 | - | - | 500, loss
500, acc | Adam | - | 50 | 64
128 | mean |
| ECC | 1
2 | 3 | 32 (cpu)
8 (gpu) | 1e-1
1e-2 | 32
64 | 1000 | - | 0.05
0.25 | 500, loss
500, acc | SGD | ECC-LR | - | - | sum |
| GIN | see hidden units | 1 | 32
128 | 1e-2 | 32 (5 layers)
64 (5 layers)
64 (2 layers)
32 (3 layers) | 1000 | - | 0
0.5 | 500, loss
500, acc | Adam | Step-LR (step: 50, gamma: 0.5) | - | - | sum |
| GraphSAGE | 3
5 | 1 | 32 (cpu)
16 (cuda) | 1e-2
1e-3
1e-4 | 32
64 | 1000 | - | - | 500, loss
500, acc | Adam | - | - | - | mean
max
sum |

