# OpenReview forum: "A Fair Comparison of Graph Neural Networks for Graph Classification"
_ICLR.cc/2020/Conference — Accept (Poster)_

### Official Review · AnonReviewer1 · 2019-10-15
**Official Blind Review #1**

**Rating:** 6

**Review:**



********** Post-Rebuttal Update **********

I appreciate that authors provided comments on all the raised concerns and updated the paper accordingly. I think the revised paper is of a higher quality.

Although the paper's experimental setup could be done in a more proper way -- nested CV; lack of which degrades the conclusiveness of the comparisons, I still think the paper is better to be accepted than not, given that 1) as pointed out by the authors the computational cost of the current experiments was quite high already, 2) the issues of replicability and reproducibility are important and GNNs are quite popular for various applications and 3) the results are interesting and important to be considered for future GNN works. So, I am happy to change my rating to weak accept. However, for the record, I would like to mention the following point:

The authors removed the word nested from the paper which is good since I still believe it does not correctly reflect what’s been done in the work. However, in the reply, they imply that several works including Varma & Simon, 2006 [1] would consider their method a nested CV. They attribute this different viewpoint to the reviewer’s misunderstanding of CV for *k-fold* CV. That is not true. CV requires “cross” testing which is absent from the inner loop of this work. I refer the interested readers to actually the mentioned paper by the authors: Varma & Simon, 2006 [1], sectio:background. Page 2 clearly defines cross validation requiring a for-loop for several evaluation across different training/test sets and page 3 defines nested CV as requiring an inner loop and  outer loop of this kind. So, as far as I understand, based on [1], the proposed method is not nested CV since the inner (model selection) part uses the classic single train/test split without any loop.


********** Summary **********

The paper conducts an empirical study of 5 recently-proposed graph neural networks (GNN). Three questions are studied:
1) For some selected set of classification tasks (chemistry and social networks) and datasets (9 different datasets), how do the considered GNN models perform, relative to each other, given the same hyperparameter search strategy and a “nested” cross validation scenario.
2) How much does the structure (graph edges) bring on top of a multi-layer perceptron (MLP) operating on merely the node features?
3) Do the considered GNN models exploit structural information of the input graphs beyond that of the node degrees?


********** Strengths and Weaknesses **********

+ it is shown that the inclusion of nodes’ degree in its feature representation brings a very large performance boost. This is very interesting.
+ it is shown that on some datasets, the results of a simple baseline -- only operating on the node features, can achieve similar results to the elaborate GNNs. This result is very informative and suggests that this baseline should always be included in GNN works.
+ some of the results in table 4 contradicts the corresponding papers which can be informative for the practitioners of the field.
+ the reproducibility and replicability problems, that is the motivation of this work, are important concerns of the field.

- reproducibility and replicability problems is not specific to research done on graph neural network and is a caveat for the general machine learning research as discussed by Lipton and Steinhardt 2018., and in fact for the current scientific practice at large as pointed in the recent NSA report [Reproducibility and Replicability in Science, 2019]. So, it may be important to raise and investigate this in different subfields of machine learning, such as GNN, individually. However, the current abstract and introduction of the paper (before the last sentence of intro’s second paragraph), associates this problem specifically to the research conducted in GNN. I strongly believe the better formulation is to refer to Lipton and Steinhardt 2018 as the troubling trend in ML as well as the NSA report in general and then mention that in the current work the authors focus on the subfield of GNN *classification*.
- page 1: “For instance, it is often unclear how hyper-parameters have been selected or which validation splits have been used.”. These being *often* the case in GNN research is not backed up by statistics and is not shown to be specific to GNN. Even if it’s assumed the 5 GNN methods , under this paper’s scrutiny, are representative of GNN classification, GNNs are used well beyond graph classification.
- it should be clearly discussed what are the additional information that this work brings on top of Shchur et al. 2018. Is it only the shift of focus from node classification to graph classification?
- the two notions of “fairness in comparison” and “ablation studies” are sometimes conflated. For instance, the argument against “fairness in comparison” of using one-hot encoding in GIN is not a matter of fairness (since it’s a novel proposal for node features of GNNs) but rather a matter of “ablation studies” (to show the effectiveness of individual components of a new GNN method).

- many concerns regarding the nested cross validation:
a) it is important to note that the paper (as described in algorithm 2) does not do “nested cross validation” despite the claim in various parts of the work. Algorithm 2 conducts model selection based on a fixed train-val split. As such, there is no “cross” validation even with the minimum of 2 folds.
b) the standard deviations reported in table 3 and 4 are unreasonably high. This important observation is not properly discussed. This high std is in contrast to the standard deviation reported in the baseline papers and makes most methods fall into one-std interval of each other. I strongly suspect that this is due to the non-nested cross validation that is done in this work. That is, the hyperparameters are found using a single val set consisting of 9% of all data. A proper nested cross validation would test each set of hyper parameters against k_in validation folds to make a robust inference of best hyperparameters.
c) there are 3 runs used “to smooth the effect of unfavourable random weight initialization on test performances”, however the 10-fold outer CV should take care of this to some extent. I would argue it would have been much more helpful to use this budget to do a proper CV for model selection using 3 folds.
d) The datasets are quite small (600-5000 samples). That might make the 10-fold cross validation problematic and overly optimistic in its performance report (and high variance). It might have been better to do say 5-fold cross validation but repeat the whole process twice. Of course, this is an old dilemma in the pratice regarding the bias and variance of the estimated CV error but as far as I am aware there is a general agreement that getting “closer” to leave-one-out setup is not a good idea, Furthermore, in my anecdotal experience 10-fold CV for a dataset with only 600 samples can really be problematic.

- would the ranking of GNN methods change with different types of the node features being used?  Some GNN setups might be better at representation learning and thus should work better on raw features (as opposed to hand engineered ones).

- how is the number of parameters (or capacity) across different GNN models and the baseline? Has this been taken into account for a fair comparison? For instance, could it be that higher/lower number of parameters explain the differences across GNNs and baseline due to over/under fitting?

********** Disclaimer. I did not thoroughly check the paper’s report of the 5 GNN methods (summarized in table 1).


********** Final Decision **********

I believe the paper has merits as well as interesting findings. It is also true that this is an important concern in machine learning research. However, there are many issues that degrades the quality of the paper. Of all those critiques, I suggest “Weak Reject” mainly due to the ones raised regarding a proper nested cross validation; which is the main proposal of the paper.



********** Minor Points **********
- Page 1: “Our results put on a fair and unique reference scale many published results which, as we document, were obtained under unclear experimental settings.“: please rephrase; at least one preposition is missing.
- Page 3: Nested Cross Validation: the brief textual explanation makes the simple idea more complicated than it is. Instead, I suggest to use citation as well as a small figure or short algorithm describing it since it’s the focus of the paper. It will also serve pedagogical purposes.
- Table 1: precisely define the two mark “A” (ambiguous) and “-” (lack of information).
- page 7: “higher or equal than” → “higher than or equal to”
- in page 6, it’s mentioned that some experiments took more than 72 hours. This sounds excessive for such small datasets. My guess is that this is due to the fact that large number of epochs are allowed (e.g. 5000 in table A.3) in conjunction with extremely small learning rate (e.g. 1e-6). The question is if those extreme hyperparameter settings are actually important for this study?
- the first paragraph of 6.1 argues that “GNNs are still unable to exploit the structure on such datasets [D&D, PROTEINS, ENZYMES]”. While this might be true, the conclusion is only based on 5 methods and limited by the design choices such as the architecture. It should be toned down.
- page 5, “Features” paragraph is not very clearly written. For instance, from the sentence: “More in detail, in the former nodes ...”, it’s hard to understand what “former” refers to.
- mean and standard deviation should have the same number of decimals (table 3,4)
- page 4: “Moreover, the authors applied early stopping, which entails the use of a validation set“. Early stopping, in a less common setup, can be used by only looking at the training set and stopping with the same n-patience strategy as used in this work.
- page 4: “we conform to the available code and do not use sampled neighborhood
Aggregation.“: needs more explanation.


********** Points of improvements **********

The main point of improvement would be a proper nested cross validation setup.
The authors can also consider the ReScience journal: https://rescience.github.io/



**Experience Assessment:**

I have published one or two papers in this area.

**Review Assessment: Checking Correctness Of Derivations And Theory:**

N/A

**Review Assessment: Checking Correctness Of Experiments:**

I carefully checked the experiments.

**Review Assessment: Thoroughness In Paper Reading:**

I read the paper thoroughly.

---

> ### Author Response · Authors · 2019-11-11
> **Authors' response to Reviewer #1**
>
> We thank the reviewer for writing this detailed review. We understand the reviewer’s concerns, and we have put our best effort to address them in the paper. Before answering each point in detail, we would like to make an important remark to place our replies into the right perspective.
> This work’s major contribution revolves around the re-evaluation of recently published results under a rigorous, reproducible and uniform framework, aimed at clearly separating the role of model selection from model assessment. This is something that is definitely lacking in the field of GNNs for graph classification, as we have shown in Section 4. The priority of this study is therefore on this specific issue.
> About the choice and quality of the statistical estimator, we argue that each method of performance assessment by resampling is subjected to trade-offs, which involve bias/variance balancement as well as computational requirements. In this respect, our choice is to keep our methodology as close as possible to the experiments performed by other works in literature.
>
> In the following comments, we provide an answer to each question posed by the reviewer.

---

> > ### Author Response · Authors · 2019-11-11
> > **Authors' response to Reviewer #1 - Major Points (part 1)**
> >
> > **QUESTION**
> > a) it is important to note that the paper (as described in algorithm 2) does not do “nested cross validation” despite the claim in various parts of the work. Algorithm 2 conducts model selection based on a fixed train-val split. As such, there is no “cross” validation even with the minimum of 2 folds.
> >
> > - Page 3: Nested Cross Validation: the brief textual explanation makes the simple idea more complicated than it is. Instead, I suggest to use citation as well as a small figure or short algorithm describing it since it’s the focus of the paper. It will also serve pedagogical purposes.
> >
> > **ANSWER**
> > Here, we think that the misunderstanding between us and the reviewer is mostly a matter of terminology. In our paper, the term “cross-validation” in “nested cross-validation” was to be interpreted in the more general sense of “estimating the accuracy on an independent test set”, rather than referring to any particular methodology of achieving this (such as k-fold cross-validation). In literature, there are cases where, even though the term nested CV is used, the associated methodology is not an outer and inner k-fold CV (see e.g. the seminal work in Varma & Simon, 2006 [1]).
> >
> > However, we agree with the reviewer that this particular terminology may confuse the reader. We also acknowledge that, in Section 3.1, we made an unclear use of such terminology. For these reasons, we have removed references to the term “nested” in the paper to avoid confusion, and we explicitly detailed our evaluation procedure; following the reviewer’s suggestion, we also refer to the Appendix for a visual explanation of the overall process. Hopefully, our description should now be unambiguous.
> >
> >
> > [1] Varma S, Simon R. Bias in error estimation when using cross-validation for model selection. BMC Bioinformatics. 2006;7:91. Published 2006 Feb 23. doi:10.1186/1471-2105-7-91.
> >
> > **QUESTION**
> > b) the standard deviations reported in table 3 and 4 are unreasonably high. This important observation is not properly discussed. This high std is in contrast to the standard deviation reported in the baseline papers and makes most methods fall into one-std interval of each other. I strongly suspect that this is due to the non-nested cross validation that is done in this work. That is, the hyperparameters are found using a single val set consisting of 9% of all data. A proper nested cross validation would test each set of hyper parameters against k_in validation folds to make a robust inference of best hyperparameters.
> >
> >
> > **ANSWER**
> > We carefully double-checked the standard deviations reported in the original papers of DGCNN, DiffPool, ECC, GIN, and GraphSAGE. To ease the discussion we summarize them as follows:
> >
> > DGCNN: repeat 10-fold CV 10 times and take the average of the 10 results. Model selection is done on a single random fold. Clearly, computing this average leads to much smaller standard deviations, and a comparison with other methods is not appropriate.
> > DiffPool: no standard deviation is reported.
> > ECC: no standard deviation is reported.
> > GIN: reports standard deviations, which have similar values as in this work.
> > GraphSAGE: standard deviations taken from GIN paper. GIN and GraphSAGE std intervals (when provided) always overlap but for NCI1. Moreover, stds have similar values as in this work.
> >
> > For these reasons, we do respectfully  not agree with the reviewer when saying that standard deviations are higher than those reported in the original papers. The reviewer is correct in saying that a small validation set may cause problems in terms of generalization estimates when doing model selection. However, any choice regarding performance estimation by resampling is related to a trade-off. For example, using an inner k-fold with small k may lead to bias-related problems at training time (since the model would leverage less training data). On the other hand, increasing k would have made the overall computational cost unsustainable.
> > To give the reviewer an idea of the computing power that was required to do this work, by assuming 1 hour of GPU computation for each experiment the amount of time required to replicate all experiments would be of 5.37 GPU years. Using (say) an internal 3-fold would have meant to scale this number by 3, making the overall cost of the evaluation roughly exceed 16 GPU years. That said, we chose this experimental setting in the first place to be as close as possible to the experimental procedures adopted in the GNNs we analyzed; in fact, all of the considered papers use 10-fold CV to assess the performance of the respective models, but they lack reproducibility in their model selection procedure (as we thoroughly discussed in Section 4).

---

> > > ### Comment · AnonReviewer1 · 2019-11-15
> > > **Post-rebuttal comments**
> > >
> > > I appreciate that authors provided comments on all the raised concerns and updated the paper accordingly. I think the revised paper is of a higher quality.
> > >
> > > Although the paper's experimental setup could be done in a more proper way -- nested CV; lack of which degrades the conclusiveness of the comparisons, I still think the paper is better to be accepted than not, given that 1) as pointed out by the authors the computational cost of the current experiments was quite high already, 2) the issues of replicability and reproducibility are important and GNNs are quite popular for various applications and 3) the results are interesting and important to be considered for future GNN works. So, I am happy to change my rating to weak accept. However, for the record, I would like to mention the following point:
> > >
> > > The authors removed the word nested from the paper which is good since I still believe it does not correctly reflect what’s been done in the work. However, in the reply, they imply that several works including Varma & Simon, 2006 [1] would consider their method a nested CV. They attribute this different viewpoint to the reviewer’s misunderstanding of CV for *k-fold* CV. That is not true. CV requires “cross” testing which is absent from the inner loop of this work. I refer the interested readers to actually the mentioned paper by the authors: Varma & Simon, 2006 [1], sectio:background. Page 2 clearly defines cross validation requiring a for-loop for several evaluation across different training/test sets and page 3 defines nested CV as requiring an inner loop and  outer loop of this kind. So, as far as I understand, based on [1], the proposed method is not nested CV since the inner (model selection) part uses the classic single train/test split without any loop.

---

> > ### Author Response · Authors · 2019-11-11
> > **Authors' response to Reviewer #1 - Major Points (part 2)**
> >
> > **QUESTION**
> > c) there are 3 runs used “to smooth the effect of unfavourable random weight initialization on test performances”, however the 10-fold outer CV should take care of this to some extent. I would argue it would have been much more helpful to use this budget to do a proper CV for model selection using 3 folds.
> >
> > **ANSWER**
> > The final training runs serve a different purpose than a 3-fold internal CV for model selection. In fact, running the selected model a number of times is necessary to loosen the dependency from the starting point of the training procedure (i.e., the random initialization of the weights). That said, running a 3-fold CV for model selection is considerably more expensive from a computational point of view, as it multiplies costs by 3 for each configuration to try, rather than training 3 more times once the best configuration is selected.
> >
> >
> > **QUESTION**
> > d) The datasets are quite small (600-5000 samples). That might make the 10-fold cross validation problematic and overly optimistic in its performance report (and high variance). It might have been better to do say 5-fold cross validation but repeat the whole process twice. Of course, this is an old dilemma in the pratice regarding the bias and variance of the estimated CV error but as far as I am aware there is a general agreement that getting “closer” to leave-one-out setup is not a good idea, Furthermore, in my anecdotal experience 10-fold CV for a dataset with only 600 samples can really be problematic.
> >
> > **ANSWER**
> > The reviewer’s concerns on datasets size and the possible high variance of the 10-fold CV performance estimator are indeed legitimate. Nonetheless, we would like to emphasize that these datasets are widely used in literature for GNN benchmarking. Thus, their adoption in this work addresses a specific design choice: to conform as much as possible to the experiments made in other works in the field.
> > As regards the choice of 10-fold CV for performance assessment, our argument is similar to the one used to answer to point b) raised by the reviewer. Other estimators have their own pros and cons (e.g. Hastie, Tibshirani & Friedman, 2009 [1]): for example, using a 5-fold CV would have made use of less data for model selection (both for train and validation), which could possibly result in less performing models and bias-related problems. As there is no “right choice” in this case, we choose the experimental setting in accordance with what was already done and also recommended in the literature.
> >
> > [1] Hastie, T., Tibshirani, R., & Friedman, J. (2009). The Elements of Statistical Learning: Data Mining, Inference, and Prediction (2nd ed.). Stanford, CA: Stanford University.
> >
> >
> > **QUESTION**
> > - would the ranking of GNN methods change with different types of the node features being used? Some GNN setups might be better at representation learning and thus should work better on raw features (as opposed to hand engineered ones).
> >
> > **ANSWER**
> > This question is valuable, and it has also been raised by other reviewers. The reason why we used uninformative features on social datasets was to check the ability of GNNs to compute better representations, and we observed that on average GIN does a good job on all datasets, a result which is also backed up by theoretical findings. Then we included the degree information to understand how much that addition helped GNNs to generalize better. While the investigation of the effect of other features is beyond the scope of this work, it is of practical importance and should be addressed in future works.
> >
> >
> > **QUESTION**
> > - how is the number of parameters (or capacity) across different GNN models and the baseline? Has this been taken into account for a fair comparison? For instance, could it be that higher/lower number of parameters explain the differences across GNNs and baseline due to over/under fitting?
> >
> > **ANSWER**
> > The configurations tried were designed to allow all GNNs to leverage similar ranges of capacity. We included the configurations reported in the original papers and explored alternatives with higher/lower capacity.
> > Even though GNNs have higher capacity than the baselines in general, we observe that an overly-parameterized baseline is not able to overfit the training data. To see this, a baseline with 10000 hidden units and no regularization is not able to overfit NCI1 training data (67% training accuracy), while GIN can easily achieve around 100% (as also demonstrated in the original paper). This indicates that structural information largely affects the ability to fit the training set (even when this does not result in better generalization on unseen graphs). We thank the reviewer for the suggestion; we have added a discussion about this in Section 6.

---

> > ### Author Response · Authors · 2019-11-11
> > **Authors' response to Reviewer #1 - Major Points (part 3)**
> >
> > **QUESTION**
> > - reproducibility and replicability problems is not specific to research done on graph neural network and is a caveat for the general machine learning research as discussed by Lipton and Steinhardt 2018., and in fact for the current scientific practice at large as pointed in the recent NSA report [Reproducibility and Replicability in Science, 2019]. So, it may be important to raise and investigate this in different subfields of machine learning, such as GNN, individually. However, the current abstract and introduction of the paper (before the last sentence of intro’s second paragraph), associates this problem specifically to the research conducted in GNN. I strongly believe the better formulation is to refer to Lipton and Steinhardt 2018 as the troubling trend in ML as well as the NSA report in general and then mention that in the current work the authors focus on the subfield of GNN *classification*.
> >
> > **ANSWER**
> > We do agree with the reviewer, these problems are not limited to GNNs and it is better to start with a more general abstract and introduction. To address the reviewer’s concern, we have included a first paragraph that introduces the reader to the more general problem of reproducibility and replicability, which has already been discussed in the suggested bibliography. The fact that our work focuses on the subfield of GNNs follows naturally from the second introductory paragraph. Finally, we extended the abstract accordingly.
> >
> >
> > **QUESTION**
> > - page 1: “For instance, it is often unclear how hyper-parameters have been selected or which validation splits have been used.”. These being *often* the case in GNN research is not backed up by statistics and is not shown to be specific to GNN. Even if it’s assumed the 5 GNN methods , under this paper’s scrutiny, are representative of GNN classification, GNNs are used well beyond graph classification.
> >
> > **ANSWER**
> > We think the reviewer is correct and it was not our intention to generalize beyond reasonable our criticism. We rephrased the sentence saying that some of the common reproducibility problems we encountered concern hyper-parameter selection and correct data splits.
> >
> >
> > **QUESTION**
> > - it should be clearly discussed what are the additional information that this work brings on top of Shchur et al. 2018. Is it only the shift of focus from node classification to graph classification?
> >
> > **ANSWER**
> > The valuable work of Shchur et al. 2018 gives two main contributions. The first is showing that the random train/validation/test split can impact on classification performance for node classification benchmarks; the second is to propose a rigorous re-evaluation of some GNNs for node classification. Similarly to Shchur et al. 2018, we share the authors’ goal of evaluating some GNNs for the task of graph classification. The crucial difference from that work, however, is that we propose a sound re-evaluation of benchmark results that, as it emerges from Section 4, are not reproducible nor uniform in the features used. In addition, we investigated the impact of the degree on performances, and we have gathered evidence that some GNNs, in the form proposed up to now, might not be exploiting the structure on some tasks. This issue could be the subject of future studies in this area, which could help better understand GNNs from an empirical point of view.
> > Thanks to the reviewer for the advice; Section 2 has now been extended accordingly.
> >
> >
> > **QUESTION**
> > - the two notions of “fairness in comparison” and “ablation studies” are sometimes conflated. For instance, the argument against “fairness in comparison” of using one-hot encoding in GIN is not a matter of fairness (since it’s a novel proposal for node features of GNNs) but rather a matter of “ablation studies” (to show the effectiveness of individual components of a new GNN method).
> >
> > **ANSWER**
> > We believe there has been a misunderstanding due to an unclear phrasing of our sentence. Our definition of “fairness in comparison” is related to the comparison of different models under the same input representations, while the way in which GIN represents the degree information is one of the possible ways to incorporate features. Hence, the two concepts are unrelated. We thank the reviewer for highlighting this, and we have rephrased our statement in the paper.

---

> > ### Author Response · Authors · 2019-11-11
> > **Authors' response to Reviewer #1 - Minor Points**
> >
> > - Page 1: “Our results put on a fair and unique reference scale many published results which, as we document, were obtained under unclear experimental settings.“: please rephrase; at least one preposition is missing.
> >
> > Thank you for the suggestion; we have rephrased the sentence.
> >
> >
> > - Page 3: Nested Cross Validation: the brief textual explanation makes the simple idea more complicated than it is. Instead, I suggest to use citation as well as a small figure or short algorithm describing it since it’s the focus of the paper. It will also serve pedagogical purposes.
> >
> > In light of the previous answers, we have already revised Section 3 by simplifying the text and citing a foundational paper on this matter. For the sake of space, we included in the Appendix a visualization of Nested Cross Validation which identifies a hold-out technique as a viable way of selecting a model in this framework.
> >
> >
> > - Table 1: precisely define the two mark “A” (ambiguous) and “-” (lack of information).
> >
> > We have revised the caption of Table 1 according to the reviewer’s suggestion.
> >
> >
> > - page 7: “higher or equal than” → “higher than or equal to”
> >
> > We have corrected this typo, and thank the reviewer for finding it out.
> >
> >
> > - in page 6, it’s mentioned that some experiments took more than 72 hours. This sounds excessive for such small datasets. My guess is that this is due to the fact that large number of epochs are allowed (e.g. 5000 in table A.3) in conjunction with extremely small learning rate (e.g. 1e-6). The question is if those extreme hyperparameter settings are actually important for this study?
> >
> > We note that the hyper-parameters the reviewer is referring to in their specific example were only used for the baselines. These, however, were the fastest models to train. The choice of such low learning rates for the baseline was especially useful on the ENZYMES dataset, which exhibited high instability during training.
> > Instead, the range of hyper-parameters used to select other models was designed to also include the proposed hyper-parameterizations in the cited papers. For instance, DiffPool is trained for 3000 epochs in the original paper, but the main training cost came from the fact that DiffPool generates dense representations of pooled graphs. On the other hand, ECC was really slow to train because it applies an MLP to each edge in the graph, and it also has the GPU-related problems reported in the original paper.
> >
> >
> > - the first paragraph of 6.1 argues that “GNNs are still unable to exploit the structure on such datasets [D&D, PROTEINS, ENZYMES]”. While this might be true, the conclusion is only based on 5 methods and limited by the design choices such as the architecture. It should be toned down.
> >
> > The reviewer is correct, we have toned down the sentence as requested.
> >
> >
> > - page 5, “Features” paragraph is not very clearly written. For instance, from the sentence: “More in detail, in the former nodes ...”, it’s hard to understand what “former” refers to.
> >
> > We have revised the content of the “Features” paragraph following the reviewer’s advice.
> >
> >
> > - mean and standard deviation should have the same number of decimals (table 3,4)
> >
> > We have applied the correction as requested.
> >
> >
> > - page 4: “Moreover, the authors applied early stopping, which entails the use of a validation set“. Early stopping, in a less common setup, can be used by only looking at the training set and stopping with the same n-patience strategy as used in this work.
> >
> > We have clarified the related sentence according to the reviewer’s recommendation.
> >
> >
> > - page 4: “we conform to the available code and do not use sampled neighborhood Aggregation.“: needs more explanation.
> >
> > We have revised the sentence accordingly.

---

### Official Review · AnonReviewer3 · 2019-10-23
**Official Blind Review #3**

**Rating:** 8

**Review:**

This paper provides an empirical comparison between several existing graph classification algorithms, aimed at providing a fair comparison among them, as well as proposing a simple baseline that does not take into account graph structural information.

Overall, I found the experimental section very thorough and sound, with proper ways of performing parameter tuning and reporting test accuracies. On the other hand, I found this paper to miss some deeper insights into why some models perform better than others, and what are the challenges provided by these graph datasets. Also, there are other interesting dimensions of comparison that are not considered (see details below), as well as insights that could benefit future work in the area.

As a disclaimer, I would like to mention that I am more familiar with graph node classification methods, as opposed to whole-graph classification (which is the focus of this paper), so I cannot assess very well the authors’ choice of which models to compare, and which datasets they were tested on.

Positive aspects of the paper:
a)  Extensive experiments, which try to find the best parameter configuration for each of these models.
b) I appreciate the addition of the structure-agnostic baselines, although I was missing major details about the particular choice of baselines (see below).
c) The writing is very clear and easy to follow.

Points where I found the paper lacking:
1.  As mentioned above, I believe there are insufficient details about the baselines. For instance what is the global sum pooling over? If it is over neighbors, then this is not exactly “structure-agnostic”. If it is over features, then why was this architecture chosen as opposed to just a multilayer perceptron, without the Deep Sets component? I see the value in an order-invariant component when aggregating features over a node and its neighbors, but why is this necessary at node level?

2.  What is the range of values considered for each parameter in the hyperparameter validation phase? For instance, I found it a bit surprising in table A.3. that all models need 500 patience.

3. I found the discussion over the results to be quite limited. For instance:
 a) The authors point out that the performance on the NCI1 dataset is different than all the others (it is the only dataset where the baseline is not the best). How is this dataset different?
b) How do these different models compare depending on certain dataset features? Perhaps a model is better than the rest if a dataset has some particular properties.
c) Are the differences in performance shown in Tables 3-4 indeed significant, given the standard deviations, or these models practically perform the same?
d) What makes these datasets challenging, such that the best performing models get up to 70-80%? This could provide interesting insights for future work.

If the authors clarify some of the issues above, especially about the results discussion part, I believe this paper may indeed be of value to the graph classification community.

**Experience Assessment:**

I have read many papers in this area.

**Review Assessment: Checking Correctness Of Derivations And Theory:**

N/A

**Review Assessment: Checking Correctness Of Experiments:**

I carefully checked the experiments.

**Review Assessment: Thoroughness In Paper Reading:**

I read the paper thoroughly.

---

> ### Author Response · Authors · 2019-11-11
> **Authors' response to Reviewer #3**
>
> We thank the reviewer, whose suggestions helped us improve the clarity and quality of the presentation. We are glad they found the experimental section very thorough and sound, which was our major priority. In the following comment, we provide an answer for each clarification asked by the reviewer, and we revised our paper accordingly.

---

> > ### Author Response · Authors · 2019-11-11
> > **Authors' response to Reviewer #3 - Points 1) and 2)**
> >
> > 1. The global sum pooling sums the features of all nodes in the graph together, and it yields a single feature vector representing the graph embedding. As a consequence, this aggregation technique of the baseline acts at graph level, not at the node level, and therefore it is structure agnostic. We hope that this clarifies the doubts of the reviewer.
> >
> > On chemical tasks, where node features are represented as one-hot vectors, summing all node features (i.e. global sum pooling) in a graph corresponds to counting how many times an atom of a certain type occurs in that graph. In other words, the sum is injective and if two graphs have different sets of nodes (regardless of the structure, which is not taken into account by our baseline) the resulting embedding will be different. These are then given to an MLP to perform graph classification.
> > On social datasets, we do not have this nice property, as the only feature we used is the degree represented as a continuous number. Therefore, we decided to use the DeepSets architecture, which is capable (in theory) of implementing all permutation-invariant functions over sets of elements, while an MLP cannot. In our scenario, the set is given by the nodes in the graph.
> >
> > We thank the reviewer for pointing this out. We have clarified this aspect in the paper to improve the quality of the presentation.
> >
> >
> > 2. In our work, we performed model selection through an exhaustive grid search. We tried all possible configurations of the hyperparameter values that are specified in Table A.3. When using early stopping there is always a trade-off between exploration of hypotheses space and computational costs; by setting a very high patience value, we chose to prioritize exploration to favor a better model selection.

---

> > ### Author Response · Authors · 2019-11-11
> > **Authors' response to Reviewer #3 - Point 3)**
> >
> >
> > 3a. We thank the reviewer to give us the opportunity to develop this point further. The results reported in Table 3 on NCI1 (but also on REDDIT and COLLAB) seem to suggest that structural information has a positive influence on GNNs performance. We would like to note that, in general, exactly quantifying the influence of the structure for a task requires domain-specific expertise, and it is therefore non-trivial to assess that; we mentioned this in Section 5. While we are able to make general observations about the whole pool of datasets, evaluating the input/target relationships on a particular dataset deserves a dedicated study.
> > We took this chance to address the point in Section 6.
> >
> > 3b. Our primary concern for this work was to provide researchers with a uniform and rigorous experimental setting (along with benchmarking results) that can be used as a reference for future studies on graph classification. For this reason, we think that a thorough study of the effect of dataset-specific features is better suited to be the object of interesting future works. Nonetheless, we provide an insight into what the answer might be for the case of degree features in social datasets, which seem to influence performances up to the point that model rankings can change drastically (Section 6). We thank the reviewer for the suggestion: we improved the manuscript by adding these considerations.
> >
> > 3c. What we have observed in this work, which is consistent with what we found in the literature (at least for models under consideration) is that standard deviations overlap sometimes significantly (IMDB) and sometimes much less so (REDDIT-MULTI and COLLAB). One of the insights of our work is indeed related to the reviewer’s question, because we show how structure-agnostic baselines can perform very similarly (in terms of overlapping std intervals) to many state-of-the-art GNNs. This result is, in our opinion, highly significant, because it shows that a rigorous evaluation leads to reconsider the relevance of these architectures in terms of representation learning. The main takeaway is that future researchers have now the chance to rigorously compare with these models by comparing their mean values and standard deviations in a unified way, in contrast with what has been often done up until now.
> >
> > 3d. A possible reason why these models cannot generalize well on unseen data may be due to noise in the dataset. Inspired by the reviewer’s comment, we performed an informal check on graph isomorphisms and the associated labels on the IMDB-BINARY dataset, which has a manageable size and no node labels (hence graph isomorphism is less expensive to check). Interestingly, we found a consistent number of pairs of isomorphic graphs, of which approximately one third are inconsistently labeled. This might explain why higher test accuracies cannot be reached. This is an important matter that needs to be extensively investigated in future works, and we sincerely thank the reviewer for the valuable suggestion. We are including the scripts needed to replicate this experiment on IMDB in the provided code. However, we would like the reviewers’ opinion on whether to include this finding in the paper, as it is a quite informal argument that would clearly need further studies.

---

### Official Review · AnonReviewer2 · 2019-10-28
**Official Blind Review #2**

**Rating:** 6

**Review:**

This type of benchmarking paper is long overdue for graph classification with deep neural networks.  The paper would've been strongly if it had the following:

1. Considered more structural features than simple node degree and clustering coefficient.  Prior work [1] has looked at such features and answered questions like: How do structural features improve classification performance?And, which structural features are the most useful?

2. Investigated which graph neural network performs better for which graph structures (preferential attachment, small world, regular, etc) and for how much homophily.

3. Investigated the robustness of graph neural networks on classification as the structure of graphs become more random (e.g., by rewiring edges while maintaining degree distribution).

[1] B. Gallagher, T. Eliassi-Rad. Leveraging Label-Independent Features for Classification in Sparsely Labeled Networks: An Empirical Study. Lecture Notes in Computer Science: Advances in Social Network Mining and Analysis, Springer, 2009.

**Experience Assessment:**

I have published in this field for several years.

**Review Assessment: Checking Correctness Of Derivations And Theory:**

I assessed the sensibility of the derivations and theory.

**Review Assessment: Checking Correctness Of Experiments:**

I carefully checked the experiments.

**Review Assessment: Thoroughness In Paper Reading:**

I read the paper thoroughly.

---

> ### Author Response · Authors · 2019-11-11
> **Authors' response to Reviewer #2**
>
> We sincerely thank the reviewer for the constructive criticism and for saying that our work has potential value. In this paper, we provide a partial answer to the reviewer’s question 1, by investigating the effect of using/not using degree information on social datasets. As regards the other two questions, we think that the proposed ideas could be interesting and valuable follow-ups of this work, whose primary goal is to provide a uniform, rigorous and reproducible re-evaluation of popular benchmarks for GNN models that all researchers can use as a reference for graph classification. Once the evaluation framework is uniform for all models, one can start to reason more effectively on the structural inductive bias of different approaches and how more sophisticated structural features affect performances.
> For completeness, we have also incorporated [1] in Section 5, as it is relevant to this work.

---

### Public Comment · ~Simon_Shaolei_Du1 · 2019-11-08
**Graph Kernels?**

Dear authors,

This is a very important paper! A fair comparison of methods for graph classification will definitely contribute a lot to the community!

However, the paper only studies GNNs but does not study any graph kernel methods. Note, unlike image data, on graph classification datasets, GNN does not always outperform graph kernels. See our recent work: https://arxiv.org/abs/1905.13192
and previous works:
https://arxiv.org/abs/1809.02670
http://jmlr.csail.mit.edu/papers/v12/shervashidze11a.html

---

> ### Author Response · Authors · 2019-11-13
> **Authors' Response to Simon Shaolei Du**
>
> Dear Simon,
>
> We are sincerely glad you share our enthusiasm for this work.
>
> We are aware of the importance of graph kernels for graph classification, and we thank you for pointing out important references, one of which is mentioned in our paper. However, the scope of our analysis is restricted to GNNs.
>
> We have also released the code and data splits to easily implement and evaluate new models in our framework. We are planning to create a public page in which to list results of methods that follow our experimental procedure. Let us know if you are interested in providing us with the results!

---

> > ### Public Comment · ~Keyulu_Xu1 · 2019-12-23
> > **Related work: GNTK is infinitely-wide GNN**
> >
> > GNTK is equivalent to infinitely wide GNNs and is not normal kernel. Thus GNTK is related work and hope to see authors incorporate into the final version!

---

### Author Response · Authors · 2019-11-11
**On a separate note**

We would like to thank the reviewers for the insightful comments; we strongly believe they have helped us greatly improve the quality of this paper.

---

### Public Comment · ~Chen_Cai1 · 2020-02-26
**One relevant paper**

Dear authors,

This is very interesting and solid work! I would like to also point out another relevant paper on benchmarking graph classification. We showed in paper [1] an extremely simple baseline based on node (and its neighbors) degree can achieve comparable results against many state-of-the-art graph kernels and graph neural networks on non-attributed graphs. It can also be seen as a variant of graph neural networks where there is no learning involved.

[1] A simple yet effective baseline for non-attributed graph classification
 https://arxiv.org/abs/1811.03508

---

### Decision · Program_Chairs · 2019-12-19

**Decision:**

Accept (Poster)

**Comment:**

The paper provides a careful, reproducible empirical comparison of 5 graph neural network models on 9 datasets for graph classification. The paper shows that baseline methods that use only node features (either counting node types, or summing node features) can be competitive. The authors also provide some guidelines for ways to improve reproducibility in empirical comparisons of graph classification.

The authors responded well to the issued raised during review, and updated the paper during the discussion period. The reviewers improved their score, and while there were reservations about the comprehensiveness of the set of experiments, they all agreed that the paper provides a solid empirical contribution to the literature.

As machine learning becomes increasingly popular, papers that perform a careful empirical survey of baselines provide an important sanity check that future work can be built upon. Therefore, this paper, while not covering all possible graph neural network questions, provides an excellent starting point for future work to extend.